# 'A double-edged sword': Perceived benefits and harms of active case-finding for people with presumptive tuberculosis and communities—A qualitative study based on expert interviews

**Olivia Biermann**[1]*, **Raina Klüppelberg**[1], **Knut Lönnroth**[1], **Kerri Viney**[1,2], **Maxine Caws**[3,4], **Salla Atkins**[1,5]

**1** Department of Global Public Health, Karolinska Institutet, Stockholm, Sweden, **2** Research School of Population Health, College of Health and Medicine, Australian National University, Canberra, Australia, **3** Department of Clinical Sciences, Liverpool School of Tropical Medicine, Liverpool, United Kingdom, **4** Birat Nepal Medical Trust, Lazimpat, Kathmandu, Nepal, **5** New Social Research and Global Health and Development, Faculty of Social Sciences, Tampere University, Tampere, Finland

* olivia.biermann@ki.se

## Abstract

### Background

Active case-finding (ACF), also referred to as community-based tuberculosis screening, is a component of the World Health Organization's End TB Strategy. ACF has potential benefits but also harms, which need to be carefully assessed when developing and implementing ACF policies. While empirical evidence on the benefits of ACF is still weak, evidence on the harms is even weaker. This study aimed to explore experts' views on the benefits and harms of ACF for people with presumptive TB and communities.

### Methods

This was an exploratory study. Semi-structured interviews were conducted with a purposive sample of 39 experts from international, non-governmental/non-profit organizations, funders, government institutions, international societies, think tanks, universities and research institutions worldwide. Framework analysis was applied.

### Results

Findings elaborated perceived benefits of ACF, including reaching vulnerable populations, reducing patient costs, helping raise awareness for tuberculosis among individuals and engaging communities, and reducing tuberculosis transmission. Perceived harms included increasing stigma and discrimination, causing false-positive diagnoses, as well as triggering other unintended consequences related to screening for tuberculosis patients, such as deportation of migrants once confirmed to have tuberculosis. Most of the perceived benefits of ACF could be linked to its objective of finding and treating persons with tuberculosis early

**Data Availability Statement:** Data cannot be shared publicly as they are highly sensitive. In view of this, as part of the consent process prior to their participation in the study, we specified that confidentiality and anonymity would be strictly maintained and that only the participating authors would have access to the data collected. With these assurances, ethical clearance was obtained from the Swedish Ethical Review Authority in Stockholm (2017/2281-31/2). For questions regarding data requests, please contact: maike.winters@ki.se.

**Funding:** OB, KL and MC are funded by the EU-Horizon 2020-funded IMPACT-TB project (grant 733174). The grant was received by MC. URL: https://ec.europa.eu/programmes/horizon2020/en. The funder had no role in study design, data collection and analysis, decision to publish, or preparation of the manuscript.

**Competing interests:** The authors have declared that no competing interests exist.

**Abbreviations:** ACF, Active Case-Finding; COREQ, COnsolidated criteria for REporting Qualitative research; TB, Tuberculosis; WHO, World Health Organization.

(theme 1), while ACF was also perceived as a "double-edged sword" and could cause harms, if inappropriately designed and implemented (theme 2). The analysis underlined the importance of considering the benefits and harms of ACF throughout the screening pathway. The study provides new insights into the perceived benefits and harms of ACF from the perspectives of experts in the field.

## Conclusion

This study highlights gaps in the evidence base surrounding ACF and can stimulate further research, debate and analysis regarding the benefits and harms of ACF to inform contextual optimization of design and implementation of ACF strategies.

## Introduction

Tuberculosis (TB) remains the world's leading infectious disease killer, even though it is curable and preventable [1]. Yearly, approximately 10 million people fall ill with TB; the estimated gap between incident and notified TB cases in the world was 2.9 million people in 2019 [1].

So-called "passive case-finding" has previously been the principal approach to TB case-finding [2]. It relies on people with signs and symptoms of TB seeking care. However, this approach is inadequate to ensure early diagnosis and treatment of all people with TB [3] and several reviews have shown that a large pool of TB patients remains undetected despite efforts to improve passive case-finding [4–8]. Moreover, nationally representative TB prevalence surveys have demonstrated that a considerable proportion of persons with undiagnosed TB disease in the community are asymptomatic or have only vague symptoms, which makes them less likely to seek care or to be correctly diagnosed [1, 9, 10].

Active TB case-finding (ACF) is defined as the systematic identification of people with presumed active TB, in a predetermined target group, using tests, examinations or other procedures that can be applied rapidly [11]. ACF is synonymous with systematic screening for active TB, although it implies screening outside of health facilities. It can offer benefits over passive case-finding, especially for those who are asymptomatic or have atypical symptoms in the early stages of disease [1, 12]. ACF is therefore a key component of the WHO End TB Strategy [13], which aligns its targets to those under Goal 3 of the Sustainable Development Goals [14]. ACF aims primarily at improving early detection and treatment of TB. ACF can also identify people who are eligible for treatment of latent TB infection by ruling out active disease and people at high risk of developing TB. Moreover, ACF can help map out risk factors and socio-economic determinants that need to be addressed to prevent TB in a given population [15].

Pre-conditions for implementing ACF are complex [11, 16, 17] and include, for example, high-quality TB diagnosis, treatment, care, management and support for patients [11]. Before embarking on ACF, it is also important to consider who may benefit from ACF. People with a heightened risk of poor treatment outcomes if diagnosis is delayed (eg people living with HIV) may derive a greater potential benefit from ACF compared to others [17]. In this regard, the WHO guidelines for systematic screening contain recommendations for screening specific risk groups for TB [11, 18]. Most of these recommendations are conditional as they depend on the available resources or setting.

Identified benefits of ACF include reduced diagnostic delays [12, 13, 19], improved TB case detection rates [20–22] and decreased patient costs prior to diagnosis [19, 23–25]. At the

community level, ACF may lead to reductions in mortality [26], transmission and prevalence of TB [27]. Meanwhile, potential harms associated with ACF comprise unintended negative effects of being correctly diagnosed, eg stigma and discrimination [12, 28], and the harms caused by a false-positive or a false-negative diagnosis [17]. While many articles reflect on the benefits and harms of ACF based on the perspectives of patients [19, 20, 23–25] and the community [21, 22], this article considers experts' perceptions. Balancing the potential benefits and harms of ACF is vital for everyone, particularly for certain groups of people such as migrants who may risk deportation if TB is diagnosed [17], employees who lack legal protection [17, 29], and people who may not have requested to participate in ACF in the first place [15].

Identifying the potential benefits and harms of a health intervention is often difficult because benefits are intended, whereas harms are unintended and cause unwanted effects [30]. One systematic review has shown that lay people and health personnel tend to overestimate the benefits of screening for health conditions and underestimate the harms [31]. In the context of ACF, a recent survey showed that National TB Programme managers often emphasize the potential benefits of ACF [32].

There is a lack of evidence on the benefits and harms of infectious disease screening in low- and middle-income settings, as much of the available evidence seems to focus on non-communicable disease in high-income countries, eg cancer [30, 33–35]. There is also a lack of evidence regarding the perceptions of benefits and harms, and how such perceptions influence infectious disease policy development and implementation, including for ACF [36].

This study is a first step towards building evidence. The aim of the study was to explore experts' perceptions of the potential benefits and harms of ACF. The study focuses on the perceptions of international ACF experts as informants. As key stakeholders with many years of experience in the field of ACF policy development and implementation, their perceptions may profoundly influence decision-making related to the development and implementation of national and global ACF policies.

## Materials and methods

This was an exploratory qualitative study based on semi-structured expert interviews [37], in line with an interpretive, descriptive approach [38]. The research team used the COREQ (COnsolidated criteria for REporting Qualitative research) Checklist [39] to report the study (S1 File). The methods have previously been described in detail in a study that was based on information from the same interviews [40]. This study has been approved by the Swedish Ethical Review Authority in Stockholm (2017/2281-31/2). Participants received background information about the study and provided written informed consent.

### Recruitment and sample selection

The interviewees were purposively sampled to include stakeholders involved in ACF policy development and implementation. The research team compiled the initial list of interviewees based on their knowledge of networks of experts and on the published scientific literature [40]. The list was discussed with, expanded and verified by two independent experts in the field. The primary investigator (OB) contacted 50 individuals via email. Of these, 39 (78%) agreed to participate, while 11 (22%) (seven of who were women) declined participation due to lack of time or interest [40]. The 39 participants were based at international, non-governmental/non-profit organizations, funders, government institutions, international societies (such as the International Society of Travel Medicine, but in the TB field), think tanks, universities and research institutions, and one independent consultant (Table 1).

**Table 1. Institutional affiliation, sex and geographical location of interviewees.**

|  | Female (n) |  | Male (n) |  | Total (n) | Total (%) |
|---|---|---|---|---|---|---|
| **Interviewees** | **7** | **18%** | **32** | **82%** | **39** | **100%** |
| **Affiliation** |  |  |  |  |  |  |
| International organization | 2 | 5.1% | 14 | 35.9% | 16 | 41.0% |
| Non-governmental/non-profit organization | 1 | 2.6% | 3 | 7.7 | 4 | 10.2% |
| Funder | 0 | 0.0% | 4 | 10.3% | 4 | 10.2% |
| International society | 1 | 2.6% | 1 | 2.6% | 2 | 5.1% |
| Think tank | 1 | 2.6% | 0 | 0.0% | 1 | 2.5% |
| Research institution | 1 | 2.6% | 2 | 5.1% | 3 | 7.6% |
| University | 1 | 2.6% | 5 | 12.8% | 6 | 15.3% |
| Government institution | 0 | 0.0% | 2 | 5.1% | 2 | 5.1% |
| Independent consultant | 0 | 0.0% | 1 | 2.6% | 1 | 2.5% |
| **Country income level based on the World Bank's classification [41]** |  |  |  |  |  |  |
| Low-income country | 1 | 2.6% | 7 | 17.9% | 8 | 20.5% |
| Lower middle-income country | 1 | 2.6% | 4 | 10.3% | 5 | 12.8% |
| Upper middle-income country | 1 | 2.6% | 1 | 2.6% | 2 | 5.1% |
| High-income country | 4 | 10.3% | 20 | 51.3% | 24 | 61.5% |

## Data collection

OB collected the data between February and May 2018 through semi-structured interviews via the telephone or face-to-face. Of the eleven face-to-face interviews, eight were conducted during a field visit to Nepal, two during WHO meetings and one at an international organization [40].

The first interview was conducted as a pilot, based on which the interview guide was adjusted (S2 File). The guide contained two sections. The first section comprised questions regarding the experience of the interviewee related to ACF projects and policy processes [40]. The second section included questions related to the personal views, values and preferences of the interviewees regarding ACF. For example, we asked about the perceived benefits and risks of ACF for the individual, the community, the health system and compared to other interventions for early case detection (S2 File). This study covers the questions raised in the second section, while the questions in the first section were analyzed independently and without overlap in a separate study [40].

After providing information about the study and obtaining informed written consent, OB conducted the interviews in English. The data collection aimed to ensure that the sample would hold adequate information power to develop new knowledge [42]. Information power implies that the more information (relevant to the study) a sample holds, the fewer participants would be needed [42]. As such, given that the sample provided a lot of relevant information and themes were judged by the team to repeat near the end of the interviews, we found the sample to hold high information power. The typical duration of an interview was 30–60 minutes. OB transcribed 10 of the audio-recorded interviews verbatim, while the rest were transcribed by a professional company [40]. We offered all participants the opportunity to view their transcripts for comments or correction, however, only three participants requested to see the transcripts. No comments or corrections were made by those who chose to view the transcripts. The anonymity and confidentiality of the interviewees were ensured by unique assigned number codes and removing all identifiers except the respondent affiliation in the presentation of the results.

**Table 2. Example of the coding process.**

| Interviewee | Quote | Code | Category | Theme |
|---|---|---|---|---|
| Interviewee 6, international organization, low-income country | "So, if you diagnose early, you treat early, so there is no other indirect cost, cost of the severe illnesses. So that is also good." | Reducing costs through early diagnosis | Cost reduction | ACF benefits are often linked to the overall ACF objective of early diagnosis and treatment |

### Data analysis

The data were analyzed using framework analysis as described by Gale et al. [43] using *ATLAS. ti* (Scientific Software Development GmbH). The data were analysed abductively; identifying themes a priori, while identifying additional themes from the data. RK and OB independently coded all interviews, resolving differences through discussion. RK developed the initial analytical framework, on which OB and SA provided feedback. RK then charted the data into a framework matrix and interpreted the data by writing memos for each of the seven identified categories. She discussed the memos with OB and SA who verified the analysis. From here, two major themes were identified. Lastly, the categories were mapped onto the TB screening pathway [30] to illustrate their occurrence at different stages of the pathway. Table 2 provides an example of the coding process.

The preliminary findings were shared at different scientific conferences, providing unique opportunities for validating the findings [40]. For the presentation of preliminary findings at the World Union Conference on Lung Health, personalized invitations were sent to all 39 interviewees. A few interviewees attended and two provided feedback. As such, the presentation of preliminary findings gave an opportunity for member-checking. No direct changes were made based on the validation and member-checking, but these processes helped to reflect on the findings critically. Furthermore, triangulation and constant discussion across different researchers from diverse backgrounds throughout the analysis helped ensure comprehensiveness and a reflexive analysis of the data [44]. Moreover, great emphasis was put on fair dealing in the presentation of the results to ensure a wide range of interviewees and their viewpoints would be represented [44] and quoted.

OB is a doctoral student in public health sciences, focusing on qualitative research and ACF. While OB conducted the interviews, she had an appropriate "distance" to the participants, not knowing any of the participants personally. She also had an equal interest in exploring both the potential benefits and harms of ACF. RK is a nurse with a master's degree in global health and experience in qualitative research. The multidisciplinary research team furthermore consisted of a medical doctor (KL), an epidemiologist (KV), a microbiologist (MC) and a social scientist (SA). KL previously worked at the World Health Organization where he coordinated the development of the systematic screening guidelines. The team's diverse background and experience helped reflect on results from different perceptions. The varied backgrounds and perceptions of ACF were balanced through careful triangulation of views and perspectives.

### Results

We identified two overarching themes in the data: 1) ACF benefits are often linked to the overall ACF objective of early diagnosis and treatment and 2) ACF-related harms are often caused by inappropriate implementation. On the one hand, the categories identified included as benefits: reaching vulnerable populations, reducing patient costs, helping raise awareness for TB among individuals and engaging communities, and reducing TB transmission. On the other hand, increased stigma and discrimination, causing false-positive diagnoses, and triggering

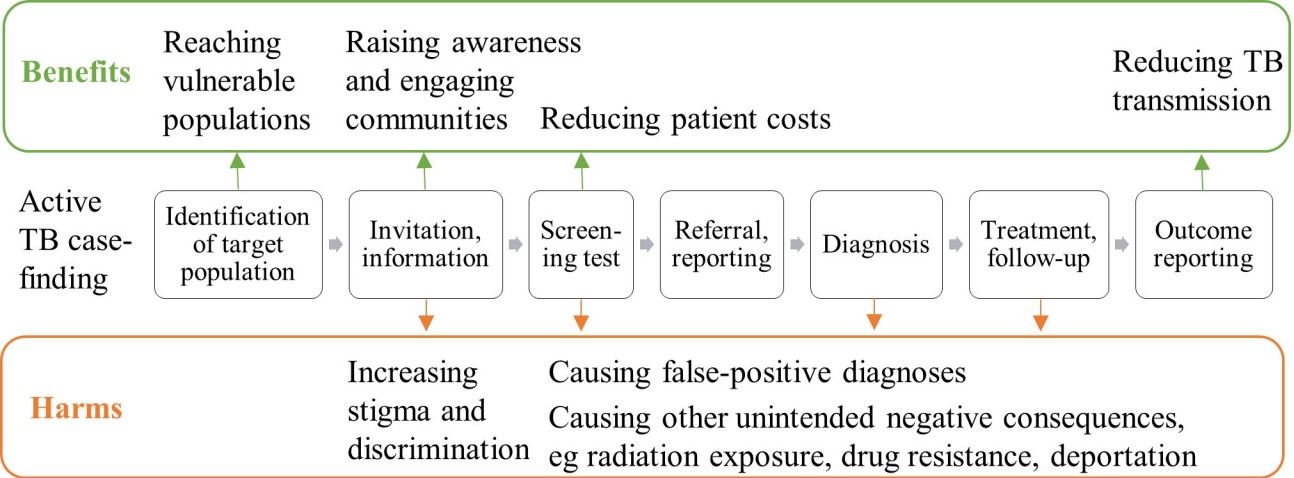

**Fig 1. Considering benefits and harms of ACF throughout the screening pathway.** This figure shows the seven-step screening pathway (adapted from WHO Europe [30]) for active tuberculosis case-finding from the identification of a target population to outcome reporting. The categories on the perceived benefits and harms of ACF are mapped along the pathway.

other unintended consequences were identified as harms. We mapped the categories on the perceived benefits and harms of ACF along the screening pathway, dividing the categories according to benefits and harms (see Fig 1). We thereby captured the importance of considering the benefits and harms at different stages of ACF to reap the potential benefits and mitigate potential harms.

## ACF benefits are linked to the overall ACF objective of early diagnosis and treatment

The theme on ACF benefits focused strongly on participants' emphasis of finding and treating TB patients early through ACF. 'Early' was perceived as diagnosing someone with TB when the disease is less advanced, the patient is less infectious and the chances to be cured are high. As such, many interviewees described the close link between early diagnosis, treatment and improved treatment outcomes, and many other benefits of ACF, which are described in the following sections. A common stance was that ACF is necessary to complement passive case-finding:

> "The benefits will vary from country to country (. . .). What we see with traditional case-finding: either there are long delays with patients accessing TB services and getting put on appropriate treatment, or [they are] missed entirely. (. . .) Passive case-finding is not really reaching as many people as it needs to, soon enough and efficiently enough."–Interviewee 13, non-profit organization, upper middle-income country

According to the interviewees, ACF could reduce TB transmission among individuals and communities screened, eg by finding TB patients early before they infect others.

Contact investigation could help identify a TB patient's household or social contacts with TB and prevent them from transmitting the disease further. Yet, the interviewees also emphasized that the reduction of transmission could be theoretical, hinting at the lack of evidence to back up these claims.

The interviewees, especially those based in low-income and lower middle-income countries, described how access to healthcare was limited in certain areas, such as in urban slums or

remote villages, eg due to lack of infrastructure or high transportation costs. The benefit of ACF was closely linked to providing services effectively to vulnerable participants. Interviewees underscored the notion that ACF could help reach vulnerable populations by overcoming access barriers to diagnostic services, eg by making free services available within communities to accommodate the needs of those people who may not be able attend screening otherwise.

> "Walking for half an hour, two hours to have a check-up is not satisfactory. It's not desirable. It's not acceptable by the clients, you know? So, in the form of ACF going out to the community has many benefits."–Interviewee, 3, government institution, low-income country

Furthermore, the benefits of ACF were seen to link with a potentially high yield of TB cases, if ACF was targeted at groups with a high risk for TB, such as people living in congregate settings or among immunocompromised individuals. One participant explained that: "A small proportion of patients may never come to the facility. That's when you need to design a way to reach them, and ACF is a way to reach them" (interviewee 30, funder, high-income country). Another interviewee mentioned inconvenient opening hours of health facilities as a major argument for doing ACF:

> "People are not willing to come to the health facilities (. . .). They are the laborers, they are the drivers, like this type of profession, and they are very busy. They totally deny coming to our health facilities because they have to work. Otherwise, they cannot run their family. That's one of the major issues. So, we have to launch these types of activities which are friendly to that target group."–Interviewee 5, government institution, low-income country

Patient costs also emerged as an important consideration when doing ACF. The interviewees had similar opinions regarding the potential of ACF to reduce patient costs by facilitating early diagnosis and treatment. Costs included transportation fees that individuals would pay to visit health facilities, charges for consultations, tests and treatments, potentially related to multiple health care visits prior to the correct diagnosis being obtained, as well as indirect costs due to income loss.

An important category that linked with finding patients early was raising awareness of TB while performing ACF. The interviewees emphasized that ACF could help raise awareness about TB and engage communities.

> "I have seen that with ACF you'll raise awareness. You sensitize the community, which will be beneficial in the long run."–Interviewee 6, international organization, low-income country

Respondents described how awareness-raising could help people recognize TB symptoms, understand the mode of transmission, understand that the disease is curable and create a demand for TB care and healthcare overall. Awareness-raising was also potentially linked to early diagnosis, as symptoms were quickly recognized. The involvement of community health workers in ACF could furthermore support awareness-raising and engagement.

> "An advantage that I see (. . .) if you put in place an ACF programme: it generates a kind of buzz around the issue. So it's also a way of advocating for the disease and making it more known and we really need that for TB because especially in our countries people (. . .) think it doesn't exist anymore."–Interviewee 17, international organization, high-income country

In terms of barriers for awareness-raising and community engagement, one participant affiliated with a government institution in a low-income country mentioned that communities

may lack trust in the government and government-supported initiatives. Another participant working for an international organization in a lower middle-income country thought that health workers' high workloads may prevent them from spending additional time on aware-ness-raising when doing ACF.

## ACF-related harms are often caused by inappropriate implementation

The interviewees described how ACF-related harms were often caused by inappropriate imple-mentation, which is not to say that well-implemented ACF would automatically neutralize all possible harms. For example, ACF may be "set up as a parallel system, which ultimately drives newly diagnosed people into the failing system" (interviewee 24, international organization, high-income country). A failing health system could be one with insufficient diagnostic and treatment capacity, interviewees explained. ACF may also not be sustainable in the long term given the higher resource requirements for ACF compared to passive case-finding, leaving "the whole government almost paralyzed to take it [ACF] on", eg once external funding ceases (interviewee 2, international organization, low-income country). Beyond funding, the inter-viewees also highlighted the potential for ACF to overburden a health system and divert resources away from other health services.

Inappropriate implementation could also cause unintended harm, in terms of increasing stigma and discrimination if people's confidentiality and privacy were not safeguarded. Conduct-ing TB screening in public spaces, workplaces or people's homes, visible to community members, co-workers, neighbors and families may stigmatize TB patients. The stigma can be in relation to TB itself, but also due to other negative associations with the disease, such as HIV infection or poverty. The lack of confidentiality and privacy in ACF was thought "to exacerbate the existing prejudice against patients with TB" (interviewee 32, university, high-income country).

> "In many communities, people wonder: 'What are the doctors or nurses doing in that house? Something is wrong.' So, there's that kind of stigma, just a blowing off people's right to keep their health problems confidential."–Interviewee 1, university, high-income country

Interviewees explained how the fear of stigma and discrimination became apparent as TB patients refused to share their address for contact-tracing, as they were afraid that future visits from health workers may disclose their TB status. Moreover, the interviewees felt that this fear could prevent patients from seeking healthcare. Patients might not collect their TB medica-tions at all, or they may prefer to travel longer distances to attend another clinic, hoping to remain anonymous. Furthermore, interviewees stressed the importance of considering stigma when implementing ACF, eg:

> "People in the community squeeze their face when we talk about TB still. [This] means there's stigma. So, that may affect individuals badly. This should be very carefully managed. You see? That's their right."–Interviewee 8, non-governmental/non-profit organization, low-income country

Yet, one interviewee stressed that stigmatization and de-stigmatization attributed to ACF are two sides of the same coin:

> "You could start getting in the argument about the stigma associated with the disease, but I could immediately flip that around and say: 'Well, one way to destigmatize it is to make it a lot more normal to expect the TB truck to be in your community and available to people

who want to be screened', and do away with the concept that this [TB] only happens to poor people."–Interviewee 21, university, high-income country

Interviewees, mostly from high-income settings, underlined that ACF may lead to false-positive diagnoses of TB, especially when screening in low-incidence and low-risk populations where ACF may cause "more harm than good" (interviewee 24, international organization, high-income country). A false-positive diagnosis was said to lead to unnecessary treatment and related side effects, costs and stigma. The interviewees described that false-positive diagnoses and related harm may occur especially in settings with suboptimal diagnostic tools:

"You knock on someone's door or you bother them while they are out in the street; then the ethical bar is significantly higher. You need to actually provide some benefit to someone and that's where screening someone, a healthy person minding their own business, is a much more ethically and technically challenging thing because you need to make sure you are not harming people by giving them false information. But, of course, a screening test is really imperfect; usually it has suboptimal sensitivity and suboptimal specificity. (. . .) The reason why I use them is they are cheap. So, it [ACF] is quite a morally fraught activity."–Interviewee 24, international organization, high-income country

Finally, interviewees mentioned potential harm for patients due to other unintended negative consequences of ACF, such as those caused by inappropriate radiation protection during X-ray screening. Moreover, interviewees described that ACF often identified asymptomatic patients who may not believe their TB diagnosis, and subsequently have limited treatment compliance and risk developing drug resistance. In the context of migrant screening, ACF may lead to deportation of migrants once confirmed to have TB, interviewees warned. Unintended consequences may also occur when ACF is conducted in a "failing system", as mentioned above. For instance, a TB patient may be identified through ACF, but the provision of TB treatment may not be ensured or may be related to direct or indirect costs for the patient. ACF may be a "double-edged sword", if poorly implemented:

"It [ACF] has two sides: If it is well designed, it can increase your case detection (. . .), reduce delays and it can reduce probably disease burden and patient cost. But if it is (. . .) poorly implemented, it can cause a lot of problems (. . .). So, it's a double-edged sword–this is a lesson learned."–Interviewee 11, international organization, high-income country

## Discussion

Our qualitative study exploring experts' views on the potential benefits and harms of ACF for presumptive TB patients and communities revealed the importance of considering the benefits and harms of ACF throughout the screening pathway to capitalize on them. On the one hand, the perceived benefits of ACF included reaching vulnerable populations, reducing patient costs, helping raise awareness for TB among individuals and engaging communities, as well as reducing TB transmission. On the other hand, the perceived harms of ACF comprised increasing stigma and discrimination, causing false-positive diagnoses, as well as triggering other unintended negative consequences. Overall, most of the benefits of ACF were tied to its objective of finding and treating persons with TB early (theme 1), while most of the perceived harms were said to be caused by inappropriate implementation of ACF (theme 2).

Many of the perceived benefits and harms of ACF are in line with the available evidence in the field, but this study also shed light on potential benefits and harms which are not yet well

documented in the literature. First, the perceived benefit that ACF reaches vulnerable populations may be reflected in studies that show reduced delays in seeking healthcare [12, 13, 19], improved TB case detection [20–22] and increased TB case notifications [19, 45–47]. Second, recent studies have shown that ACF reduces patient costs [19, 23–25], which matches with expert perceptions and seems particularly important given that between 19% and 83% of TB patient face catastrophic costs when accessing TB care, which means that they spend 20% or more of their annual household income on TB care [1]. Project Axshya in India systematically assessed the impact of ACF and advocacy, communication and social mobilisation at the individual and community levels when compared to passive case-finding. The study found improved case notification rates, reduced diagnostic delay, and reduced patient costs [19]. Third, the perceived benefit of ACF in reducing TB transmission is supported by modelling studies [48–51] and two recently published cluster randomized controlled trials [26, 27]. One trial showed an increase of case detection as a result of inviting household contacts to attend a clinic for symptom screening, physical examination and chest radiography, and found reductions in all-cause mortality [26]. The other trial evaluated the effectiveness of community-wide screening compared to passive case-finding and demonstrated reduced prevalence of TB [27]. Finally, the perceived harm of increasing stigma and discrimination has also been discussed in the literature [12, 28]. Although, empirical evidence remains weak, potentially because it may be difficult to attribute increased or decreased stigma and discrimination to a single intervention. In addition to these topics already highlighted in the literature, the interviewees noted potential benefits and harms which have a limited evidence base, such as that ACF can help raise awareness of TB among individuals and engage communities, and that false-positive diagnoses and other unintended negative consequences due to ACF cause harm.

The concerns about the benefits and harms of ACF for TB appear to be similar to those related to screening for other health conditions [33, 52–56]. However, the harms that have been described in other screening initiatives may be more closely related to the use of a certain diagnostic tool and related health outcomes, while our study showed that the perceived harms of ACF occur throughout the screening pathway. The perceived benefits of ACF that have been observed in other types of screening include awareness-raising, which, in the context of cervical cancer, include increased participation in screening [52] and, in the context of leprosy, a shortened the delay between the onset of disease and diagnosis [31]. Moreover, reduced transmission has been described as a potential benefit of malaria screening [54]. The perceived harms have also been described in screening studies for other health conditions, eg false-positive diagnoses and related psychological consequences have been well-documented in the context of cancer screening [30, 33, 57], and medical complications which have been described in atrial fibrillation screening where people had a false-positive result [55]. Moreover, a systematic review documented poor linkages to care for patients screened positive for non-communicable diseases in sub-Saharan Africa [56]. Apart from the similarities mentioned, screening may also have opposite effects, eg decreasing inequities by reaching vulnerable populations was a perceived benefit of ACF, while the literature on cancer screening has described that screening may increase inequities because people from higher socio-economic groups are more likely to participate in screening than those with a low socio-economic status [58–60].

This study identified *what* the perceived benefits and harms of ACF are and *when* they may appear along the screening pathway, leading to considerations about *how* to balance the benefits and harms in a specific context. Interviewees from different contexts had a variety of perspectives on the potential benefits and harms of ACF. For instance, it seemed that most interviewees who raised concerns about false-positive diagnoses were based in high-income countries, whereas concerns about the ability to link TB patients to treatment and care was mainly voiced by interviewees based in low- and middle-income countries. Such contextual

considerations may help ensure that ACF is well-designed and implemented. The interviewees expressed support for ACF, if the design and implementation is appropriate to minimise harm, while maximising the benefits. This is aligned to a utilitarian perspective, where the implementation of a screening programme would be justified if its benefits outweigh its harms at a reasonable cost [60]. However, it may be challenging to quantify certain factors (eg increased stigma) and draw direct comparisons between the benefits and harms (eg reduced transmission versus false-positive results). Evaluating the potential benefits and harms of ACF may be even more challenging in low- and middle-income settings, as many do not have established processes for the development of screening policies in place [61].

Overall, our findings suggest that benefits and harms of ACF must be considered throughout the screening pathway. This is, at its core, about the ethical conduct of ACF. Conducting ACF ethically means that there is a system to ensure that TB patients are linked to treatment and care to avoid unintended negative consequences of ACF, as interviewees described. This finding echoes the statement in 1968 by Wilson and Jungner [16] that the ability to treat the condition adequately when discovered is *"perhaps the most important"* of all screening criteria. Special attention should therefore be paid to that criterion, given that the availability of the most accurate (and often most expensive) screening and diagnostic tests as well as treatment varies greatly across different settings [15]. Furthermore, part and parcel of the ethical conduct of ACF are confidentiality, privacy and informed consent. The necessity of informed consent has been signaled by WHO [11], while it is also important to consider that informed consent does not remove responsibility for any harm caused through screening [62]. The understanding and promotion of informed consent may be facilitated by training health workers in communicating the potential benefits and harms of screening [30]. Marmot and colleagues [35] state, and it corresponds with the findings of this study, that *"clear communication of these harms and benefits (. . .) is of utmost importance and goes to the heart of how a modern health system should function."*

## Future research

This study identified a range of perceived potential benefits and harms of ACF, some of which have not been studied in detail. Research is needed to understand how perceptions on benefits and harms influence ACF policy development and implementation, how these perceptions differ between settings, and to better quantify the benefits and harms of ACF. This quantification will require a standardization of measurements of the benefits and harms, which could complement existing frameworks for balancing the benefits, harms, costs, values and preferences, such as the Evidence to Decision frameworks [63], to support decision-makers. In addition, high quality evidence is needed on the benefits and harms of ACF for the health system, including the cost of screening and alternative costs, eg of possible de-prioritization of passive case-finding or overall health system strengthening. Research should also explore the perspectives of health care workers and TB patients, particularly in low- and middle-income countries, and evaluate whether or how ACF contributes to raising awareness about TB and decreasing TB-related stigma.

## Strengths and limitations

This study involved a large number and diverse range of experts, key stakeholders in ACF policy development and implementation. Collectively, the interviewees have had many years of experience in the field of ACF, which, together with the member-checking carried out with selected interviewees, increase the study's trustworthiness, including its confirmability and transferability [64]. However, the transferability of the findings of this study may still be

limited given that only a minority of the interviewees were from low- and middle-income countries (38% of the interviewees) [40]. Nevertheless, all had working experience in low-and middle-income countries. Seven of the interviews with experts from low- and middle-income countries were conducted with experts from Nepal. Though all of them have different affiliations, their perspectives may be overrepresented. The results may furthermore be limited as an even smaller minority were women (18% of the interviewees) [40]. The gender bias reflects the lack of gender parity in leadership positions in the field of global health [65]. At the time of the data collection in 2018, there may have been a lack of equal representation in terms of gender and country income level in high-level committees and working groups, and some of these committees informed our sampling base. Recognizing this lack of balance in recruitment, we paid careful attention to patterns in the findings in terms of country income level or gender and highlighted the affiliations of interviewees quoted. We did not find systematic differences in perceptions according to country classification or gender. Where differences were noted we included these in the results.

## Conclusion

This study provides new insights into the perceived benefits and harms of ACF for presumptive TB patients and communities from the perspectives of experts in the field. Most of the perceived benefits of ACF could be linked to its objective of finding and treating persons with TB early, while ACF was also perceived as a '*double-edged sword*' and could cause harms if inappropriately designed and implemented. The analysis underlined the importance of considering the benefits and harms of ACF throughout the screening pathway and the need for a deeper understanding of the actual benefits and harms of ACF. We hope this study will stimulate further research, debate and analysis of the benefits and harms of ACF, building a basis upon which better policies and practices can be built.

## Supporting information

**S1 File. COREQ checklist.**
(PDF)

**S2 File. Interview guide.**
(DOCX)

## Acknowledgments

First and foremost, the authors thank the interviewees who generously shared their time to participate in the study. The authors also thank Anna Borgström, writing instructor at Karolinska Institutet University Library, for her valuable feedback in writing this manuscript.

## Author Contributions

**Conceptualization:** Olivia Biermann, Knut Lönnroth, Kerri Viney, Maxine Caws.

**Data curation:** Olivia Biermann, Salla Atkins.

**Formal analysis:** Olivia Biermann, Raina Klüppelberg, Salla Atkins.

**Funding acquisition:** Maxine Caws.

**Investigation:** Olivia Biermann, Raina Klüppelberg, Salla Atkins.

**Methodology:** Olivia Biermann, Kerri Viney.

**Project administration:** Olivia Biermann.

**Resources:** Knut Lönnroth.

**Software:** Olivia Biermann, Raina Klüppelberg.

**Supervision:** Knut Lönnroth, Kerri Viney, Maxine Caws, Salla Atkins.

**Validation:** Olivia Biermann, Raina Klüppelberg, Knut Lönnroth, Kerri Viney, Salla Atkins.

**Visualization:** Olivia Biermann, Raina Klüppelberg.

**Writing – original draft:** Olivia Biermann, Raina Klüppelberg.

**Writing – review & editing:** Olivia Biermann, Knut Lönnroth, Kerri Viney, Maxine Caws, Salla Atkins.

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
