## [Decision Letter · Decision Letter 0]

1 Oct 2020

PONE-D-20-20600

‘A double-edged sword’: Perceived benefits and harms of active case-finding for people with presumptive tuberculosis and communities – a qualitative study based on expert interviews

PLOS ONE

Dear Dr. Biermann,

Thank you for submitting your manuscript to PLOS ONE. After careful consideration, we feel that it has merit but does not fully meet PLOS ONE’s publication criteria as it currently stands. Therefore, we invite you to submit a revised version of the manuscript that addresses the points raised during the review process.

Three experts in the field reviewed your manuscript and have several comments that need to be addressed.

We look forward to receiving your revised manuscript.

Kind regards,

Susan Hepp

Academic Editor

PLOS ONE

Journal Requirements:

Reviewers' comments:

Reviewer's Responses to Questions

**Comments to the Author**

1. Is the manuscript technically sound, and do the data support the conclusions?

Reviewer #1: Partly

Reviewer #2: Yes

Reviewer #3: Yes

2. Has the statistical analysis been performed appropriately and rigorously? 

Reviewer #1: N/A

Reviewer #2: N/A

Reviewer #3: N/A

3. Have the authors made all data underlying the findings in their manuscript fully available?

Reviewer #1: Yes

Reviewer #2: No

Reviewer #3: Yes

4. Is the manuscript presented in an intelligible fashion and written in standard English?

Reviewer #1: Yes

Reviewer #2: Yes

Reviewer #3: Yes

5. Review Comments to the Author

Reviewer #1: This is an important paper highlighting both benefits and harms for Active Case Finding, in the current context where most program managers would emphasize the benefits and often not bringing up the harms. The paper in general is well written. However there are a few points which the authors need to address before this manuscript can be considered for publication, i.e.:

1. Please describe the Qualitative approach (e.g., ethnography, grounded theory, case study, phenomenology, narrative research) and guiding theory if appropriate; identifying as well the research paradigm (e.g., postpositivist, constructivist/interpretivist) if possible

2. Please elaborate the researchers’ characteristics that may influence the research, including personal attributes, qualifications/experience, relationship with participants, assumptions, and/or presuppositions; potential or actual interaction between researchers’ characteristics and the research questions, approach, methods, results, and/or transferability

3. Please elaborate the criteria for deciding when no further sampling was necessary (i.e. sampling saturation);

4. The authors reported efforts for member checking, which in the end didn’t seem to enhance trustworthiness substantially. Thus, please confirm whether the researchers have adequately employed other techniques to enhance trustworthiness and credibility of data analysis (e.g., audit trail, triangulation);

5. The biggest disappointment was that the characteristics of the experts were highly skewed to male from high income countries. This could have been prevented early on during recruitment as there are actually enough experts from low- and middle- income countries, with richer insights from the ground. In view of this limitation, I would suggest to at least present a matrix which allows examination whether there are variations of themes between experts from High Income vs Low- and Middle Income Countries.

Reviewer #2: 1. The conclusions are drawn from the data.

2. Statistical analysis is not applicable as this is a qualitative research.

3. Data has not been shared citing ethical issues. However, anonymised data, removing the remarks that could have pointed to the participants' identity, could have been shared.

4. Manuscript is well written. But some of the concerns are as follows.

1. Aim of the study is not very clear. Whether the aim is to understand the perceptions of the international experts regarding ACF and how they influence policy or as claimed the first step in exploring the actual benefits and harms of ACF. If the attempt is to find the benefits as well as the harms of ACF, the participants should have been chosen from a wider range of stakeholders. All participants in this study were involved in ACF policy development and implementation. Therefore their opinions are likely to be influenced by their professional responsibilities.

2. The impact of ACF depends a lot on the context. Stigma, discrimination and the ability of the health systems to support activities like ACF may be country specific. Therefore it would have been better if the experts were asked to opine on the benefits and harms of ACF done in specific settings rather than in general.

3. In the figure,

- 'false positive' is shown as the harm coming from diagnosis. False positives are results of the screening tests which are not basis for initiation of treatment. Confirmed diagnosis is done as per the diagnostic algorithms followed in the National TB programmes of the countries after referral reporting. After diagnosis, all positives are supposed to be true positives;

- instead of writing 'unintended negative consequences', which is non-specific, it is better to write the specific codes that emerged from the interviews. Figures should be self-explanatory.

4. In response to the point asking whether the transcripts were returned to the participants for comments or corrections in the COREQ checklist, it is mentioned as page no.7. However, in the manuscript, it is stated that all participants were invited to the conference where the findings were presented, but only few attended. The transcripts should have been shared with all participants for validation.

5. Overall, the paper is a novel attempt to throw light on the unintended harms that may occur with ACF, but what is lacking is the context, which would determine both the benefits and harm.

Reviewer #3: Summary

This is a qualitative study consisting of semi-structured interviews were conducted with a purposive sample of 39 experts from various organisations and institutions worldwide. The analysis underlined the importance of considering the benefits and harms of ACF throughout the screening pathway. The study provides new insights into the perceived benefits and harms of ACF from the perspectives of experts in the field. The study was able to provide a nice summary of all the benefits of ACF to communities and participants. This study highlights gaps in the evidence base surrounding ACF which are important and need to be taken into account when active case finding programmes are implemented. I think this work is relevant and adds to the very scanty literature in this field.

Major comments

ACF-related harms are clearly articulated and seem well thought through however I am not sure that the theme around inappropriate implementation being the cause is substantiated enough in the text. It seems that even if done well, there are still significant challenges and harms that can be caused.

The discussion is done very nicely where the expert impressions are substantiated with literature from the field.

Discussion is a bit long – may consider reducing it a bit.

Minor comments

Abstract – First sentence in the results is far too long and difficult to follow – please consider splitting it into shorter sentences.

Introduction, Line 83: “It” should be replaced with “ACF” to be clearer

Introduction, Line 84: remove “help”

Introduction, Line 85-86: make this clearer by splitting the sentence?

Introduction , Lines 89-91: please rephrase as sentence is vague and unclear

Introduction, Line 96: missing word “the”

Introduction, Line 98 – 105: the introduction does not clearly state how this study is different to what is already reported in the literature. Are these findings from patients or community members? Would be good to differentiate from this study to show how this study adds value.

Introduction, Line 109: split into two words “laypeople”

Table 1 – I wonder if it is not possible to streamline the organisations a bit – not sure International institution and international society or non-profit organisation for instance are different types of institutions.

Table 2 – start on a new page

Methods, Page 10 Line 187: word missing before “conferences’’

Results, Page 10, Line 200: include “as benefits’’ after “included’’

Discussion, Page 18, Line 378: perceived is repeated twice in the sentence

Discussion, Line 395: “and” is missing

Discussion, Line 397 – 399: rather emphasise the mortality reduction case detection by placing that at the end of the sentence.

6. PLOS authors have the option to publish the peer review history of their article (what does this mean?). If published, this will include your full peer review and any attached files.

Reviewer #1: No

Reviewer #2: **Yes: **Sonali Sarkar

Reviewer #3: **Yes: **Salome Charalambous

---

## [Author Response · Author response to Decision Letter 0]

29 Oct 2020

Point-by-point reviewers’ comments and authors’ answers

Title of the manuscript: “‘A double-edged sword’: Perceived benefits and harms of active case-finding for people with presumptive tuberculosis and communities – a qualitative study based on expert interviews” 

Reviewer 1: anonymous

1. Please describe the Qualitative approach (e.g., ethnography, grounded theory, case study, phenomenology, narrative research) and guiding theory if appropriate; identifying the research paradigm as well (e.g., postpositivist, constructivist/ interpretivist) if possible

We have described the qualitative research as well as the research paradigm accordingly on p. 6 (lines 131-132): “This was an exploratory qualitative study based on semi-structured expert interviews [34], in line with an interpretive, descriptive approach [35].”

As a reference for the interpretive, descriptive approach we have taken, we added the following reference: Thorne S, Reimer Kirkham, O’Flynn-Magee K. The analytical challenge in interpretive description. International Journal of Qualitative Methods. 2004;3(1).

2. Please elaborate the researchers’ characteristics that may influence the research, including personal attributes, qualifications/experience, relationship with participants, assumptions, and/or presuppositions; potential or actual

interaction between researchers’ characteristics and the research questions, approach, methods, results, and/or transferability 

We have elaborated on this in p. 11 (lines 212-222): “OB is a doctoral student in public health sciences, focusing on qualitative research and ACF. While OB conducted the interviews, she had an appropriate “distance” to the participants, not knowing any of the participants personally. She also had an equal interest in exploring both the potential benefits and harms of ACF. RK is a nurse with a master’s degree in global health and experience in qualitative research. The multidisciplinary research team furthermore consisted of a medical doctor (KL), an epidemiologist (KV), a microbiologist (MC) and a social scientist (SA). KL previously worked at the World Health Organization where he coordinated the development of the systematic screening guidelines. The team’s diverse background and experience helped reflect on results from different perceptions. The varied backgrounds and perceptions of ACF were balanced through careful triangulation of views and perspectives.”

3. Please elaborate the criteria for deciding when no further sampling was necessary (i.e. sampling saturation) 

We have elaborated more on the concept of information power, which we applied instead of the concept of saturation, on p. 9 (lines 172-176): “Information power implies that the more information (relevant to the study) a sample holds, the fewer participants would be needed [39]. As such, given that the sample provided a lot of relevant information and themes were judged by the team to repeat near the end of the interviews, we found the sample to hold high information power.”

4. The authors reported efforts for member checking, which in the end didn’t seem to enhance trustworthiness substantially. Thus, please confirm whether the researchers have adequately employed other techniques to enhance trustworthiness and credibility of data analysis (e.g., audit trail, triangulation) 

We clarified how we additionally triangulated the data on p. 10, lines 206-210: “Furthermore, triangulation and constant discussion across different researchers from diverse backgrounds throughout the analysis helped ensure comprehensiveness and a reflexive analysis of the data [41]. Moreover, we put great emphasis on fair dealing in the presentation of the results to ensure a wide range of interviewees and their viewpoints would be represented [41] and quoted.” 

We have also added a new reference on validity in qualitative research: Mays N, Pope C. Qualitative research in health care. Assessing quality in qualitative research. BMJ. 2000 Jan 1;320(7226):50-2.

5. The biggest disappointment was that the characteristics of the experts were highly skewed to male from high income countries. This could have been prevented early on during recruitment as there are actually enough experts from low and middle- income countries, with richer insights from the ground. In view of this limitation, I would suggest to at least present a matrix which allows examination whether there are variations of themes between experts from High Income vs Low- and Middle-Income Countries.

We fully agree that unequal representation is a major limitation of our study; it is something we should have considered much more strongly from the beginning, and something we will make sure to account for in any future studies. In terms of the gender skewedness, we had contacted seven additional women who declined participation (in addition to three men), which we have now added to p. 7, line 146. Moreover, given that our participants were experts working in leadership positions, we wonder whether the unequal representation of gender at least partly reflects the unequal representation in global health leadership and therefore our sampling space. We found at the time of data collection (2018) that the committees we approached for sampling were unbalanced in terms of country and gender representation, an issue that we are now glad to see is improving.

We appreciate the suggestion of including a matrix. Early in our analysis we realized that our sample did not equally represent males versus females, and settings with different income levels. We therefore paid careful attention to examining different themes for representation from countries with different income categories. However, we did not find a clear pattern of themes between low- and middle-income country versus high-income country, or male versus female respondents in this examination. Instead, opinions seemed to be located across settings, which makes including a matrix less useful or, we fear, even counterproductive for the presentation of our findings (quantifying results instead of presenting ideas qualitatively). 

We had elaborated on the above issues accordingly in the limitations section and have expanded our elaboration on p. 23-24 (lines 518-532): “The transferability of the findings of this study may still be limited given that only a minority of the interviewees were from low- and middle-income countries (38% of the interviewees). Nevertheless, all had working experience in low-and middle-income countries. Seven of the interviews with experts from low- and middle-income countries were conducted with experts from Nepal. Though all of them have different affiliations, their perspectives may be overrepresented. The results may furthermore be limited as an even smaller minority were women (18% of the interviewees). The gender bias reflects the lack of gender parity in leadership positions in the field of global health [62]. At the time of the data collection in 2018 there may have been a lack of equal representation in terms of gender and country income level in high-level committees and working groups, and some of these committees formed our sampling base. Recognizing this lack of balance in recruitment we paid careful attention to patterns in the findings in terms of country income level or gender and highlighted the affiliations of interviewees quoted. We did not find systematic differences in perceptions according to country classification or gender. Where differences were noted we included these in the results”

Reviewer 2: Sonali Sarkar

6. Data has not been shared citing ethical issues. However, anonymised data, removing the remarks that could have pointed to the participants' identity, could have been shared. All participants are well-known in the field and easily identifiable. 

Given that all participants of this study are high-level experts working in leadership positions, they are very well-known in the field of tuberculosis and active case-finding. It may be possible to identify them not only be references they make to organizations they work with or countries they work in, but by the way the express themselves. When signing the consent form, we promised all interviewees absolute confidentiality and anonymity, which may be compromised by making transcripts available. 

7. Aim of the study is not very clear. Whether the aim is to understand the perceptions of the international experts regarding ACF and how they influence policy or as claimed the first step in exploring the actual benefits and harms of ACF. If the attempt is to find the benefits as well as the harms of ACF, the participants should have been chosen from a wider range of stakeholders. All participants in this study were involved in ACF policy development and implementation. Therefore, their opinions are likely to be influenced by their professional responsibilities. 

We have revised the text accordingly to clearly state the aim on p. 6 (lines 122-124): “The aim of the study was to explore experts’ perceptions of the potential benefits and harms of ACF. The study focuses on the perceptions of international ACF experts as informants.” 

We agree that exploring the actual benefits and harms of ACF would require further studies and should involve a wide range of participants. 

With regards to interviewees’ professional responsibilities influencing their opinions, this is indeed likely. However, we do not perceive this as negative, as long as the interviewees expressed their opinions and not the opinions of their institutions. To ensure the latter, we highlighted to interviewees that we were interested in their personal views and often emphasized this while conducting the interviewees through prompts such as: “in your opinion…”, “what do you think…” or “what is your most important lesson learned”. 

8. The impact of ACF depends a lot on the context. Stigma, discrimination and the ability of the health systems to support activities like ACF may be country specific. Therefore, it would have been better if the experts were asked to opine on the benefits and harms of ACF done in specific settings rather than in general.

We absolutely agree that the benefits and harms of ACF are context dependent. By interviewing experts, we hoped to map a broad range of benefits and harms, illuminating them from a global perspective. Meanwhile we hope that future studies will dig deeper into the benefits and harms of ACF in specific contexts. 

We had highlighted the need for these important questions to be addressed under Future Research on p. 22-23 (lines 499-510): “Research is needed to understand how perceptions on benefits and harms influence ACF policy development and implementation, how these perceptions differ between settings, and to better quantify the benefits and harms of ACF. This quantification will require a standardization of measurements of the benefits and harms, which could complement existing frameworks for balancing the benefits, harms, costs, values and preferences to support decision-makers, such as the Evidence to Decision frameworks [60]. In addition, high quality evidence is needed on the benefits and harms of ACF for the health system, including the cost of screening and alternative costs, eg of possible de-prioritization of passive case-finding or overall health system strengthening. Research should also explore the perspectives of health care workers and TB patients, particularly in low- and middle-income countries, and evaluate whether or how ACF contributes to raising awareness about TB and decreasing TB-related stigma.”

9. In the figure, - 'false positive' is shown as the harm coming from diagnosis. False positives are results of the screening tests which are not basis for initiation of treatment. Confirmed diagnosis is done as per the diagnostic algorithms followed in the National TB programmes of the countries after referral reporting. After diagnosis, all positives are supposed to be true positives; instead of writing 'unintended negative consequences', which is non-specific, it is better to write the specific codes that emerged from the interviews. Figures should be self-explanatory.

We have revised the figure accordingly by clarifying that the screening test as well as the diagnostic test may be false-positive. We have also specified examples of what we mean with “unintended consequences”, i.e. radiation exposure, drug resistance and deportation. 

10. In response to the point asking whether the transcripts were returned to the participants for comments or corrections in the COREQ checklist, it is mentioned as page no.7. However, in the manuscript, it is stated that all participants were invited to the conference where the findings were presented, but only few attended. The transcripts should have been shared with all participants for validation.

We have now added an explicit sentence on p. 9 (lines 178-181) and corrected the page number accordingly in the COREQ checklist: ”We offered all participants the opportunity to view their transcripts for comments or correction, however, only three participants requested to see the transcripts. No comments or corrections were made by those who chose to view the transcripts.” 

11. Overall, the paper is a novel attempt to throw light on the unintended harms that may occur with ACF, but what is lacking is the context, which would determine both the benefits and harm. 

As mentioned under comment 8, we fully agree that the benefits and harms of ACF are context dependent, while we hoped to map a broad range of benefits and harms, illuminating them from a global perspective. We hope that future studies will explore the benefits and harms of ACF in specific contexts.

Reviewer 3: Salome Charalambous

12. ACF-related harms are clearly articulated and seem well thought through however I am not sure that the theme around inappropriate implementation being the cause is substantiated enough in the text. It seems that even if done well, there are still significant challenges and harms that can be caused. 

To clarify that even if done well, there could still be significant challenges and harms related to ACF, we have revised the text accordingly on p. 15 (lines 322-325): “The interviewees described how ACF-related harms were often caused by inappropriate implementation, which is not to say that well-implemented ACF would automatically neutralize all possible harms.” 

In the text, we had given examples of inappropriate implementation and linked those to the harms that we described. Examples of possible ways of inappropriately implementing ACF include that ACF “may be ‘set up as a parallel system’” (line 326), that it may be implemented despite “insufficient diagnostic and treatment capacity” (line 329), that it may “overburden a health system and divert resources away from other health services” (line 334) or that it could be implemented without safeguarding people’s confidentiality (line 336). 

13. The discussion is done very nicely where the expert impressions are substantiated with literature from the field. Discussion is a bit long – may consider reducing it a bit. 

We have slightly shortened the discussion section as suggested. We have taken out the following phrase as it was redundant to considerations we already had included under Future Research: “This highlights the need for further research in this area to document and quantify the potential harms of ACF for TB, to determine if perceptions of harm are justified and to allow an evaluation of the benefits against the harms in different contexts.” (p. 19, lines 413-416)

14. Abstract – First sentence in the results is far too long and difficult to follow – please consider splitting it into shorter sentences. 

We have revised the first sentence accordingly, breaking it up into two: “Findings elaborated perceived benefits of ACF, including reaching vulnerable populations, reducing patient costs, helping raise awareness for tuberculosis among individuals and engaging communities, and reducing tuberculosis transmission. Perceived harms included increasing stigma and discrimination, causing false-positive diagnoses, as well as triggering other unintended consequences related to screening for tuberculosis patients, such as deportation of migrants once confirmed to have tuberculosis.”

15. Introduction, Line 83: “It” should be replaced with “ACF” to be clearer

We have amended the sentence accordingly. 

16. Introduction, Line 84: remove “help” 

We have removed “help” as proposed. 

17. Introduction, Line 85-86: make this clearer by splitting the sentence?

We have split this sentence as suggested on p. 4-5 (lines 83-86): “ACF aims primarily at improving early detection and treatment of TB. ACF can also identify people who are eligible for treatment of latent TB infection by ruling out active disease and people at high risk of developing TB. Moreover, ACF can help map out risk factors and socio-economic determinants that need to be addressed to prevent TB in a given population [15].”

18. Introduction , Lines 89-91: please rephrase as sentence is vague and unclear

We have rephrased the sentence as follows on p. 5 (lines 89-91): “Pre-conditions for implementing ACF are complex [11, 16-17] and include, for example, high-quality TB diagnosis, treatment, care, management and support for patients [11].”

19. Introduction, Line 96: missing word “the” 

We have corrected the sentence accordingly. 

20. Introduction, Line 98 – 105: the introduction does not clearly state how this study is different to what is already reported in the literature. Are these findings from patients or community members? Would be good to differentiate from this study to show how this study adds value.

On p. 5 (lines 102-104), we have highlighted our study’s added value: “While many articles reflect on the benefits and harms of ACF based on the perspectives of patients [18-19, 22-24] and the community [20, 21], this article considers the experts’ perceptions.”

21. Introduction, Line 109: split into two words “laypeople” 

Have made the correction as suggested. 

22. Table 1 – I wonder if it is not possible to streamline the organisations a bit – not sure International institution and international society or non-profit organisation for instance are different types of institutions.

We have streamlined the table by merging non-governmental organizations and non-profit organizations. Moreover, we have clarified what is meant with “international societies” on p. 7 (lines 148-149): “international societies (such as the International Society of Travel Medicine, but in the TB field)”. 

23. Table 2 – start on a new page 

After making our revisions, Table 2 was not divided onto two pages anymore. 

24. Methods, Page 10 Line 187: word missing before “conferences’’

We have corrected the sentence as pointed out. 

25. Results, Page 10, Line 200: include “as benefits’’ after “included’’

We have revised the sentence accordingly. 

26. Discussion, Page 18, Line 378: perceived is repeated twice in the sentence

We have re-written the sentence as follows on p. 19 (lines 400-401): “…while most of the perceived harms were said to be caused by inappropriate implementation of ACF (theme 2).”

27. Discussion, Line 395: “and” is missing 

We have added this as suggested. 

28. Discussion, Line 397 – 399: rather emphasise the mortality reduction case detection by placing that at the end of the sentence. 

We have rephrased the sentence as follows on p. 19 (lines 419-422): “The trial showed an increase of case detection as a result of repeated household visits, symptom screening, physical examination and chest radiography, and found reductions in all-cause mortality [47].”

---

## [Editor Report · Decision Letter 1]

30 Dec 2020

PONE-D-20-20600R1

‘A double-edged sword’: Perceived benefits and harms of active case-finding for people with presumptive tuberculosis and communities – a qualitative study based on expert interviews

PLOS ONE

Dear Dr. Biermann,

Thank you for submitting your manuscript to PLOS ONE. After careful consideration, we feel that it has merit but does not fully meet PLOS ONE’s publication criteria as it currently stands. Therefore, we invite you to submit a revised version of the manuscript that addresses the points raised during the review process.

Thank you for responding to the comments of the reviewers. The manuscript has been revised as per the suggestions from the reviewers. However, some errors in citing of references need correction, for example,

1. 'The methods have previously been described in detail in a study that was based on information from the same interviews' has been cited as ref. no 36, which should be ref no. 37.

2. 'Country income level based on the World Bank’s classification' cited as 36, should be 38.

Another clarification sought is regarding the questions asked to the participants of this study. Was there any reason for asking about the benefits to the individual and risks at the community level? Both the benefits and risks mentioned in the results pertain to the individual as well as the community level.

We look forward to receiving your revised manuscript.

Kind regards,

Sonali Sarkar

Academic Editor

PLOS ONE

---

## [Author Response · Author response to Decision Letter 1]

7 Jan 2021

Point-by-point reviewers’ comments and authors’ answers

Title of the manuscript: “‘A double-edged sword’: Perceived benefits and harms of active case-finding for people with presumptive tuberculosis and communities – a qualitative study based on expert interviews” 

Reviewer comment: 'The methods have previously been described in detail in a study that was based on information from the same interviews' has been cited as ref. no 36, which should be ref no. 37. 'Country income level based on the World Bank’s classification' cited as 36, should be 38.

Authors’ answer: Thank you very much. We have corrected the referencing as indicated. In addition, we have double-checked all references for their correctness and updated/added three references: 

1. WHO. Global TB Report 2020. Geneva: World Health Organization; 2020.

The WHO Global TB Report that we originally included was from 2019. We have now cited the latest Report from 2020. Based on the new Report, we updated the estimated gap between incident and notified TB cases in the world to “2.9 million people in 2020” (instead of 3 million) (p. 4, lines 64-65) and “19% and 83% of TB patients face catastrophic costs” (instead of 27% and 83%) (p. 20, line 422). 

2. WHO. Rapid communication on systematic screening for tuberculosis. Geneva: World Health Organization; 2020.

As WHO is in the process of revising their guidelines on systematic screening, the Organization published a rapid communication that was relevant to include. We added the reference (number 18) here: “In this regard, the WHO guidelines for systematic screening contain recommendations for screening specific risk groups for TB [11, 18].”

3. Marks GB, Nguyen NV, Nguyen PTB, Nguyen TA, Nguyen HB, Tran KH, et al. Community-wide Screening for Tuberculosis in a High-Prevalence Setting. N Engl J Med. 2019;381(14):1347-1357.

The trial by Marks et al (2019) is a key reference showing benefits of ACF at the community level. We have now ensured mentioning this reference in the introduction, along another key reference (Fox et al 2018) which we had only mentioned in the discussion section: “At the community level, ACF may lead to reductions in mortality [26], transmission and prevalence of TB [27].” (p. 5, lines 100-101). 

Moreover, we have made slight revisions to the discussion section to reference not only the study by Fox et al, but also by Marks et al: “One trial showed an increase of case detection as a result of inviting household contacts to attend a clinic for symptom screening, physical examination and chest radiography, and found reductions in all-cause mortality [26]. The other trial evaluated the effectiveness of community-wide screening compared to passive case-finding and demonstrated reduced prevalence of TB [27].”

Reviewer comment: Another clarification sought is regarding the questions asked to the participants of this study. Was there any reason for asking about the benefits to the individual and risks at the community level? Both the benefits and risks mentioned in the results pertain to the individual as well as the community level. 

Authors’ answer: Thank you for this observation. We would like to clarify that we did ask about both the benefits and the risks at the individual and community level. We realized that the example questions we had included in the text were misleading. We have therefore rephrased the text as follows: “For example, we asked about the perceived benefits and risks of ACF for the individual, the community, the health system and compared to other interventions for early case detection (S2 File) (instead of: “What are the benefits of ACF in your view, at the level of the individual?”, “What are the risks of ACF from your opinion, at the level of communities?””) (p. 8, lines 168-172)

---

## [Editor Report · Decision Letter 2]

10 Feb 2021

‘A double-edged sword’: Perceived benefits and harms of active case-finding for people with presumptive tuberculosis and communities – a qualitative study based on expert interviews

PONE-D-20-20600R2

Dear Dr. Biermann,

We’re pleased to inform you that your manuscript has been judged scientifically suitable for publication and will be formally accepted for publication once it meets all outstanding technical requirements.

Modifications have been done adequately. The manuscript is ready for publication. A minor language correction needed in lines 214-215 that was missed out, can be edited.

Kind regards,

Sonali Sarkar

Guest Editor

PLOS ONE
---

## [Editor Report · Acceptance letter]

23 Feb 2021

PONE-D-20-20600R2 

‘A double-edged sword’: Perceived benefits and harms of active case-finding for people with presumptive tuberculosis and communities – a qualitative study based on expert interviews 

Dear Dr. Biermann:

I'm pleased to inform you that your manuscript has been deemed suitable for publication in PLOS ONE. Congratulations! Your manuscript is now with our production department. 

Kind regards, 

on behalf of

Dr. Sonali Sarkar 

Guest Editor

PLOS ONE